# ACTNet: A Dual-Attention Adapter with a CNN-Transformer Network for the Semantic Segmentation of Remote Sensing Imagery

**Zheng Zhang, Fanchen Liu, Changan Liu, Qing Tian * and Hongquan Qu**

School of Information, North China University of Technology, Beijing 100144, China
* Correspondence: tianqing@ncut.edu.cn

**Abstract:** In recent years, the application of semantic segmentation methods based on the remote sensing of images has become increasingly prevalent across a diverse range of domains, including but not limited to forest detection, water body detection, urban rail transportation planning, and building extraction. With the incorporation of the Transformer model into computer vision, the efficacy and accuracy of these algorithms have been significantly enhanced. Nevertheless, the Transformer model's high computational complexity and dependence on a pre-training weight of large datasets leads to a slow convergence during the training for remote sensing segmentation tasks. Motivated by the success of the adapter module in the field of natural language processing, this paper presents a novel adapter module (ResAttn) for improving the model training speed for remote sensing segmentation. The ResAttn adopts a dual-attention structure in order to capture the interdependencies between sets of features, thereby improving its global modeling capabilities, and introduces a Swin Transformer-like down-sampling method to reduce information loss and retain the original architecture while reducing the resolution. In addition, the existing Transformer model is limited in its ability to capture local high-frequency information, which can lead to an inadequate extraction of edge and texture features. To address these issues, this paper proposes a Local Feature Extractor (LFE) module, which is based on a convolutional neural network (CNN), and incorporates multi-scale feature extraction and residual structure to effectively overcome this limitation. Further, a mask-based segmentation method is employed and a residual-enhanced deformable attention block (Deformer Block) is incorporated to improve the small target segmentation accuracy. Finally, a sufficient number of experiments were performed on the ISPRS Potsdam datasets. The experimental results demonstrate the superior performance of the model described in this paper.

**Keywords:** remote sensing; semantic segmentation; transformer; adapter

## 1. Introduction

With the development of modern remote sensing technology and the launch of a series of important high-resolution remote sensing satellites, high-resolution remote sensing (HRRS) images are increasingly captured and applied to research. They contain a rich amount of information on the texture, shape, structure, and neighborhood relationship of various features. The traditional mathematical theory-based semantic segmentation methods [1–3] for the remote sensing of images can be used for relatively simple contents, but are often not suitable for images with complex features. With the excellent image feature extraction capability shown by CNN in recent years, an end-to-end network structure has been established for use in image classification, semantic segmentation, object detection, and other fields, and is effectively used for remote sensing applications [4–6].

Transformer is an architecture proposed in 2017 in the field of NLP, and is a structure for learning global features through a self-attention mechanism. It has achieved extraordinary results in the field of NLP and was quickly introduced by researchers into the field of CV. The Vision Transformer (ViT) [7] cuts images into patches and maps them

onto one-dimensional vectors for processing so that they can be converted into sequences for input into the self-attention module, which better captures long-range features and global information. ViT are slightly more accurate than CNN structures after pre-training on large-scale datasets, which demonstrates the powerful potential of Transformer in the imaging domain. Subsequently, more and more CV tasks use Transformer-based models, including semantic segmentation [8–10], target detection [11–13], pose estimation [14,15], etc.

However, the existing models still have serious shortcomings for remote sensing using multi-objective segmentation. First of all, using weights pre-trained o a large dataset to initialize the parameters leads to a better model performance [16], but Transformer lacks inductive biases in CNN, such as translation invariance and local relations, resulting in a poor performance of Transformer-based networks on small datasets. Secondly, based on the square linear relationship between the computational complexity and the image size in the ViT model, it cannot achieve better convergence in training. Thirdly, the increase in the resolution of remote sensing images brings greater intra-class differences and inter-class similarity, and while Transformer can construct a global semantic representation of the images, it loses much detailed information in the process of patching, which is particularly significant for HRRS images. In addition, the flattening process also destroys the structural information of the images, resulting in small targets or multi-branch targets with obvious texture features in HRRS images that cannot be well segmented.

To solve the above problems, this paper proposes a multi-objective segmentation network (ACTNet) with a hybrid CNN and Transformer, which is based on the Swin Transformer and uses a shifted window-based attention algorithm, so that the computational complexity is linear with the image size. In order to not change the structure of the Swin Transformer, the ResAttn module is designed as an adapter in this paper. Its dual attention mechanism ensures that sufficient global information is obtained during the training for remote sensing segmentation tasks and does not lead to excessive computation. Meanwhile, for small and multi-branch targets, we also propose a CNN-based multi-scale feature extraction module (LFE), which refers to the ResNet [17] and mainly consists of a series of convolution and pooling layers to extract as many local details of different targets as possible. In addition, a residual structure is added to the Mask2Former [18] algorithm, so that the mask feature can incorporate more information on deep-level features to improve the segmentation performance of the multi-target.

The main contributions of the article are summarized as follows:

In order to solve the problem of excessive computational complexity in the training phase of HRRS image semantic segmentation, we propose an adapter module (ResAttn) capable of remote sensing semantic segmentation. It uses a dual-attention mechanism to ensure that sufficient global information can be obtained from the feature map. For better integration into the Swin Transformer structure, we use the same patch merging method for down-sampling.

In order to enhance small target segmentation, we explore a CNN-based multi-scale feature extraction module (LFE), which aims to fully extract the texture, color, and other shallow features according to the convolutional filter weights. Meanwhile, local correlation and kernel weight sharing help to keep the parameters relatively small, which also compensates for the lack of local information extraction in Transformer.

We use a mask-based segmentation method with enhanced residual structure. The segmentation accuracy of the model on the occluded targets is improved by using residual connections to process the feature maps before and after through the multi-scale deformable attention layer.

The remainder of this paper is organized as follows. Section 2 introduces the related work. Section 3 presents the design details of our proposed network. Section 4 provides the relevant experiments and setups, and Section 5 summarizes our approach and presents the outlook for future research.

## 2. Related Work

This section describes the related work in CNN-based remote sensing semantic segmentation methods, Vision Transformer, and adapters. Table 1 and Figure 1 show the context of this section.

**Table 1.** Summary of related work.

| HRRS Image Segmentation Methods | | | Adapter |
|---|---|---|---|
| **Transformer Based** | **CNN-Based** | **CNN-Transformer** | |
| Swin [19], ST-UNet [20] | FCN [21] | CCTNet [22] | K-Adapter [23] |
| DC-Swin [24] | U-Net [25] | Swin + SASPP + SE [26] | Clip-Adapter [27] |
| TransRoadNet [28] | DeepLab [29–31] | | AdapterFusion [32] |
| SwinSUNet [33] | MC-FCN [34] | CTNet [35] | ViT-Adapter [36] |

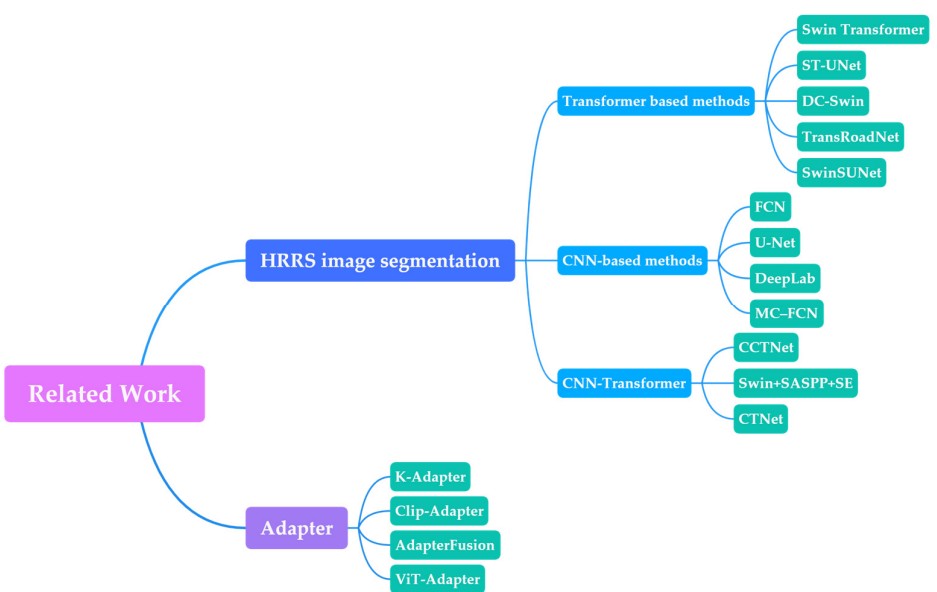

**Figure 1.** The overall context of related work.

### 2.1. CNN-Based Semantic Segmentation Methods on Remote Sensing Images

With the development of remote sensing technology and the outstanding performance of CNN in deep learning, the research related to remote sensing semantic segmentation has received wide attention. Since the introduction of the fully convolutional network (FCN), an encoder–decoder architecture has been widely used. The encoder performs convolution and down-sampling on the image to extract the image features, while the decoder recovers the spatial resolution by upsampling the small-size feature map. Based on FCN, Ronneberger et al. [25] developed the U-Net network with a symmetric encoder–decoder structure (i.e., contracting path and expansive path), where the encoder features are introduced in the decoding stage to gather more spatial information. The MC-FCN network proposed by Wu et al. [34] added a residual structure and multi-scale subconstraints based on the U-Net to improve performance in building segmentation.

Despite the successful application of Deep Convolutional Neural Networks (DCNNs) to various tasks, they lack an effective method to acquire global information, which is a critical limitation for understanding complex scenes. To address this issue, PSPNet, proposed by Zhao et al. [37], invokes the spatial pyramid pooling (SPP) method to obtain multi-scale features by pooling layers of different sizes, and then performs feature fusion and upsampling to improve the network's ability to acquire global information. Furthermore, the DeepLab model (DeepLab v2, DeepLab v3, and DeepLab v3+) proposed by Chen et al. [29–31] replaces the pooling layer in SPP with inflated convolution, allowing for the

learning of more feature information from the previous input. Although the above methods have improved the performance of CNN model segmentation, they still lack the capability to effectively extract dense target segmentation and fine-branch segmentation.

### 2.2. Vision Transformer

Due to the excellent performance of Transformer in the field of NLP, it was soon adopted by CV and presented in the Vision Transformer architecture, which relies on its attention mechanism to learn the long-distance information in images. The Swin Transformer proposed by Liu et al. [19] uses shifted window-based attention mechanisms, whose computational complexity is squarely related to the window size and linearly related to the image size. The shifted windows scheme ensures the information interaction among the windows, which enables the Transformer-based model to further explore the features in HRRS images. He et al. [20] introduced the Swin Transformer module into the U-Net shape model, which enhances the spatial feature analysis and small-scale object extraction to improve the global modeling capability. Wang et al. [24] designed the DCFAM module based on an attention mechanism and inflated convolution in the decoder to strengthen the relationship between spatial-wise and channel-wise. To improve the road extraction, Yang et al. [28] performed contextual modeling on high-level features to enhance the foreground information learning capability in order to combat similarity and occlusion problems. Zhang et al. [33] designed a Siamese U-shaped network using Swin Transformer blocks; the encoder generates multiscale features by using a hierarchical Swin Transformer.

The Transformer structure used for the extraction of global information can effectively compensate for the lack of CNN models; therefore, many researches have begun to explore suitable methods to fuse these two components. Wang et al. [22] proposed LAFM and CAFM to efficiently fuse the dual-branch features of the CNN and Transformer models. Zhang et al. [26] used depthwise-separable, convolution-based, atrous spatial pyramid-pooling modules to connect the Swin Transformer-based backbone and CNN-based decoder to capture multi-scale contextual information. The CTNet proposed by Deng et al. [35] uses a dual-stream structure to combine the Transformer and CNN models in its overall architecture, and uses concatenated semantic features and structural features to predict the scene categories.

The Introduction of the Transformer module made the remote sensing segmentation task pay full attention to the information of the target context, resulting in both improved continuity and noise immunity. To solve the problem of the high computational complexity of the attention algorithm, some attention-limited networks, such as cSwin Transformer [38], have been proposed to further reduce the computational effort, but this has led to the loss of the extraction of global features. Moreover, the networks which have Transformer as the backbone still require a large number of computational resources for transfer learning, which has a great deal of room for improvement for the training of remote sensing segmentation as a downstream task.

### 2.3. Adapters

CV tasks, such as classification, target detection, and semantic segmentation, have been significantly improved with better architectural design and large-scale high-quality datasets. However, collecting datasets for each task is too costly for the scale. To address this problem, the "pretrain-finetune" paradigm, in which large-scale datasets such as ImageNet are pre-trained to obtain weights and then applied to various downstream tasks and finetune, has been widely adopted in the CV field [16,39].

The Adapter was first proposed in NLP (Houlsby et al. [40]), and has been widely used in both the NLP and CV fields [27,32]. Its core idea is to update only the parameters in the adapter module while keeping the other parameters unchanged, so that it can achieve the same effect as finetuning. The K-Adapter structure proposed by Wang et al. [23] makes the adapter more modular for knowledge-intensive tasks. Recently, the ViT-Adapter model proposed by Chen et al. [36] successfully applied this idea to image dense prediction, where

the missing local continuity information from the ViT is supplemented by the adapter, allowing it to perform well in dense segmentation. However, it is still a challenge to design an effective adapter module to cope with multi-scale targets and dense targets in remote sensing segmentation.

## 3. Methodology

In this paper, we propose a new semantic segmentation scheme for remote sensing images. First, we will introduce the various modules contained in ACTNet's encoder and decoder and the general flow. Then we propose an adapter module (ResAttn) based on a dual attention structure to fully extract global information without excessively increasing the parameters. To enhance the model for the extraction of shallow features, such as texture, color, etc., we propose a CNN-based LFE module. Finally, we propose the Deform Block with residual enhancement to improve the segmentation of occluded targets.

### 3.1. Overall Architecture

The general overview structure of ACTNet is shown in Figure 2. The network is divided into encoder and decoder parts. In the encoder part, a stem block is used to preprocess the H $\times$ W size input image first. It consists of four convolutional layers and one pooling layer, each followed by a batch normalization and a ReLU activation function, with an output size of 1/4 of the original image. The output of the stem block is used as input for the LFE module, ResAttn module, and Swin Transformer backbone.

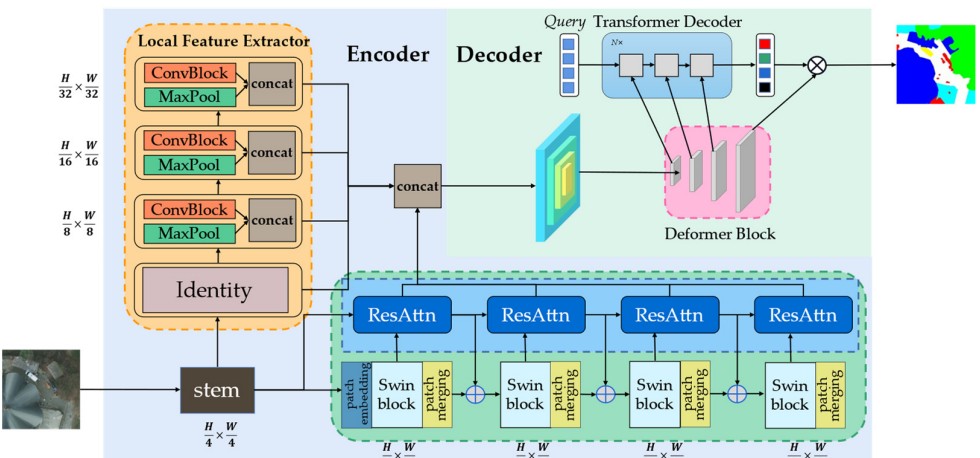

**Figure 2.** Overview structure of ACTNet.

As shown in the green area in Figure 2, due to the high resolution of the HRRS images, global modeling at large imaging sizes is required. Therefore, the Swin Transformer backbone is used as the main global modeling method, which significantly reduces the computational effort with the help of the shifted window attention algorithm. It consists of four Swin Transformer blocks, each of which contains several MSA and SW-MSA blocks in a series to form a structure.

As shown in the blue-dashed part of the green area in Figure 2, a lightweight ResAttn module is applied behind each Swin Transformer block. It has an input consisting of the output of the current Swin Transformer block and the output of the previous ResAttn module. After generating tokens and fusing them with each other, the global dependency between the features of the two levels can be derived by using the self-attention mechanism to minimize feature loss during down-sampling. In order to keep the structure of the Swin Transformer, element-wise additions are made between the output and the original Swin Transformer block output, so that the pre-trained weight information can be fully utilized during migration training. We use the same patch merging method as the Swin

Transformer uses for down-sampling the output features of the first three blocks to form a multi-scale feature extraction.

Meanwhile, we use several sets of CNN structures to obtain the low-level features of the image, as shown in the orange area of Figure 2. The feature maps obtained from the stem are fed into the LFE module; here the design pattern of the ResNet is invoked, which consists of three stages with a series of convolutions and poolings. The Identity module will not perform any processing, so that the module will output feature maps of 1/4, 1/8, 1/16, and 1/32 of the original image size, finally concatenating with the output of the 4 ResAttn modules as the encoder part of the output.

In the decoder section, we use Mask2Former as a base structure. To improve the segmentation of small objects, a multi-scale decoder structure is used, where the encoder output will be sent into the deformer block first to generate pyramid-like features. As shown in the pink area in Figure 2, here we calculate the correlation of each pixel in the feature map with the surrounding sampled points using 3 N deformable attention for 1/8-, 1/16-, and 1/32-size feature maps. We then upsample the 1/16- and 1/32-size feature maps with the 1/4-size feature map using a bilinear interpolation method for element-wise addition, in order to enhance the effect of small target segmentation while preventing network degradation. These features are next sent to the Transformer decoder module, where N-length queries with random initialization parameters will be learned to obtain global information from masked attention. After that, the mask result and the classification result are calculated with the feature map of 1/4 the original image size. Finally, the mask output and classification output are combined to obtain the network output.

### 3.2. ResAttn

As shown in Figure 2, the first ResAttn module begins with the output of the stem block. The stem block consists of 4 convolutional layers and 1 pooling layer and, as shown in Figure 3, it performs simple feature extraction and down-sampling operations on the input image, which is used to initially reduce the image size and decrease the network complexity.

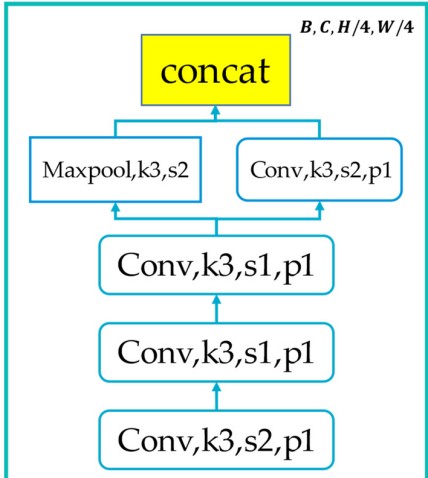

**Figure 3.** Illustration of stem block.

The existing models still have a risk of gradient disappearance as the network layer deepens, and the deep feature map loses a large number of small object features. Therefore, we propose the ResAttn module, as shown in Figure 4, which is based on an attention structure and incorporates the idea of residual structure. Specifically, it uses the output of the current Swin Transformer block and the previous ResAttn module, then generates $1 \times 1$-size tokens and fuses the features together for input into the self-attention module, which uses a multi-head self-attention algorithm. It then concatenates the two parts of the output. Finally, the result is passed through the FFN module. It contains 2 linear layers

and 2 activation functions. For this reason, the computation performed in the self-attention mechanism is mainly matrix multiplication, i.e., it is a linear transformation; therefore, its learning ability is still not as strong as the nonlinear transformation, so the expression ability of the query is enhanced by means of activation functions. The features from this step are collected as part of the encoder output. In order to keep the structure of the Swin Transformer, the same method of down-sampling is used to process the features as they are, doubling the number of channels and halving the size. The output features perform element-wise addition with the Swin Transformer block output.

Assuming that the input feature size is $(c, h, w)$, the two inputs are passed through the convolution layer to generate a token of size $(c, 1, 1)$ and a query of size $c, h \times w$, respectively. Then they are fused with each other and sent to the self-attention layer after the addition of position encoding to calculate the weight between the elements in the query sequence. For this we use the self-attention algorithm from ViT, which essentially uses a matrix multiplication to calculate the relationship between each patch and the other patches in the query, and to update the weight matrix by back propagation, whose specific formulas are as follows:

$$Attention(Q, K, V) = Softmax\left(\frac{QK^T}{\sqrt{d_k}}\right)V \tag{1}$$

$$Q = X \times W_q \tag{2}$$

$$K = X \times W_k \tag{3}$$

$$V = X \times W_v \tag{4}$$

where $X$ is the query, $W_q$, $W_k$, and $W_v$ are the learnable weight matrices, and the association between the previous layer features and the current features is constructed by self-attention. The output query is then restored to its original size and concatenated, for which we use a $1 \times 1$ convolutional layer to adjust the length to $(c, h \times w)$ and an FFN module to enhance its nonlinear representation. Finally, the image size and number of channels are adjusted by patch merging.

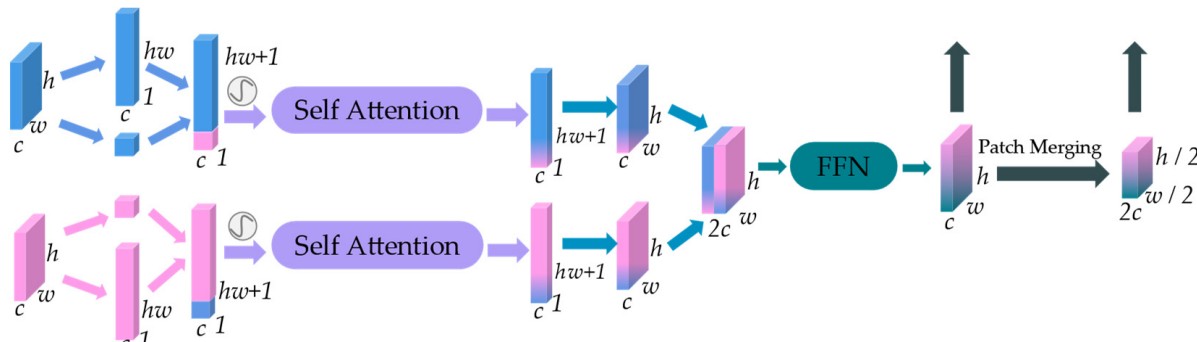

**Figure 4.** Illustration of ResAttn module.

In ACTNet we add a ResAttn module after each Swin Transformer block, and the final feature sizes obtained are 1/4, 1/8, 1/16, and 1/32 of the original image. This method achieves a result very similar to that of the feature pyramid of the SPP network.

### 3.3. LFE

Previous studies have shown that the overuse of the Transformer model in the encoder part causes the network to become less capable of extracting shallow features. This indicates there are difficulties in the extraction of most objects with distinct boundaries for multi-target semantic segmentation in HRRS images. CNN-based networks, on the other hand, can obtain local features with relatively small numbers of parameters by gradually

increasing the perceptual field through layer-by-layer convolutions, which have distinct geometric properties and are often concerned with consistency or covariance under transformations such as translation, rotation, etc. For example, a CNN convolution filter detects key points, object boundaries, and other basic units that constitute visual elements and that should be transformed simultaneously under spatial transformations, such as translation. CNN networks are a natural choice for dealing with such covariance, so that positional transformations under the same objects have little effect. Therefore, a multi-scale CNN-based LFE module is proposed to enhance the extraction and analysis of high-frequency information in images and to improve the segmentation accuracy of small and multi-branch targets to compensate for the shortcomings of the Transformer networks.

The purple area shown in Figure 5 is the LFE module, which borrows the design pattern of ResNet. We take the original image as the input, and an initial feature block of 1/4 the size of the original image is generated by stem for initial processing. Then a 3-stage convolution block is used to extract the image features. Each stage contains one maxpooling layer and one ConvBlock. Each ConvBlock has N residual convolutional blocks, as shown on the right side of Figure 5. The small cell composed of convolutional layers and residual structures ensures feature extraction while preventing network degradation. After concatenating the output of the maxpool and ConvBlock as the next input, the LFE module finally extracts the features from the original image sizes of 1/4, 1/8, 1/16, and 1/32, as the complement of the Transformer structure. The number of residual convolutional blocks of each ConvBlock in ACTNet is 3, 4, and 3, respectively, so that the number of parameters are small.

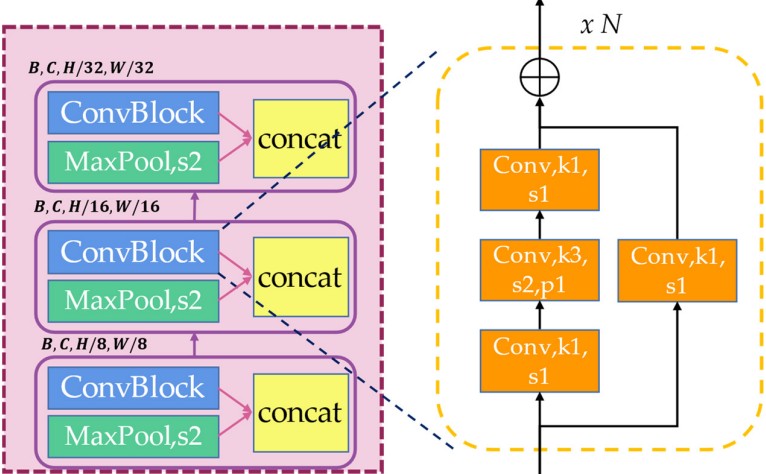

**Figure 5.** Illustration of LFE module.

### 3.4. Deformer Block and Loss Function

The decoder section can be seen in the upper right corner of Figure 2, which consists mainly of a deformer block and a Transformer decoder. After the output from the encoder module, a mask-based classification method is used for segmentation instead of the per-pixel classification that we had been using. Many objects in remote sensing images have occlusions, such as houses occluded by tree branches and cars occluded by leaves, which leads to the wrong classification of pixels. Mask segmentation predicts the class of an object using a binary mask, which works better in cases where per-pixel classification fails due to background noise or image complexity, and requires fewer parameters and computations [41].

As shown in the Figure 6, the 4-scale feature maps output by the encoder are first fed into the deformer block module. We calculate the weights using 3 N multi-scale deformable Transformers for the offset of the reference points, which are generated by each query in the

feature map sizes of 1/8, 1/16, and 1/32, where N represents 2. The deformable attention formula is as follows:

$$\text{Deformable Attention } (z_q, p_q, x) = \sum_{m=1}^{M} W_m \left[ \sum_{k=1}^{K} A_{mqk} \cdot W'_m x(p_q + \Delta p_{mqk}) \right] \tag{5}$$

where $z_q$ is obtained from the input $x$ by linear transformation, $p_q$ is a 2D vector representing the coordinates, $M$ represents the attention head, $K$ represents the number of positions sampled by 1 query in 1 head, $A_{mqk}$ represents the normalized attention weight, $W'_m x$ is the transformation matrix of value, and $\Delta p_{mqk}$ is the sampling offset.

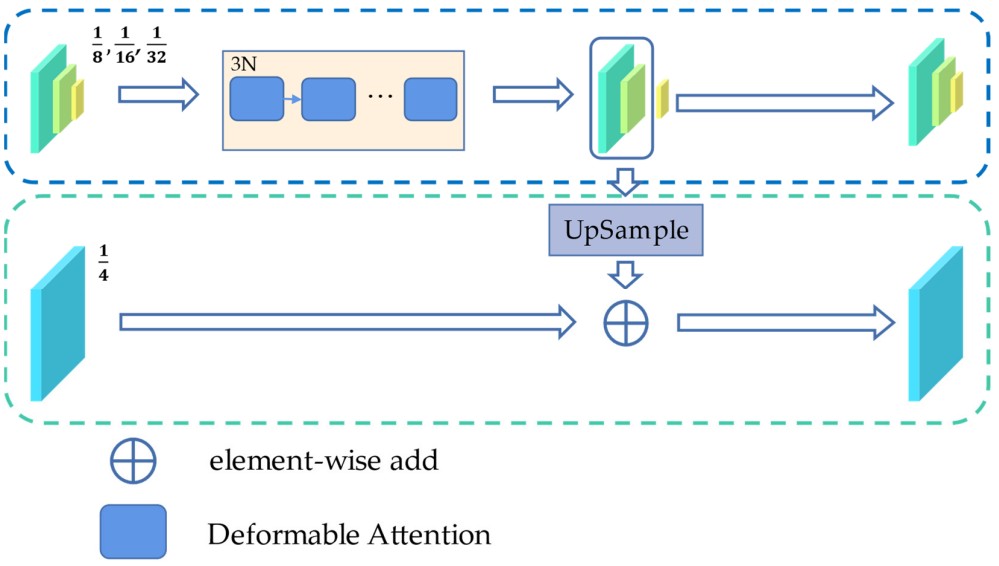

**Figure 6.** Illustration of our Deformer Block module.

Then the output of the 1/16- and 1/8-size features are added to the 1/4 feature map using bilinear interpolation upsampling to obtain the masked attention. After that, the 1/8-, 1/16-, and 1/32-size features are fed into the Transformer decoder with 3L attention blocks. Finally, the binary mask of each feature map and its corresponding classification result are calculated by query.

In order to accurately calculate the deviation between the result and the ground truth value, the loss function we use combines Cross Entropy Loss (*CELoss*), *FocalLoss*, and *DiceLoss*, each of which has its own role in improving the overall performance. The function can be expressed as follows:

$$Loss = (CELoss + FocalLoss) + DiceLoss \tag{6}$$

*CELoss* is used to calculate the category probability loss, which is suitable for multi-category tasks and is good for remote sensing multi-target segmentation. The formula is as follows, where $M$ represents the number of categories, $y_c$ is the ground truth value, and $p_c$ is the predicted value:

$$CELoss = -\sum_{c=1}^{M} y_c \log(p_c) \tag{7}$$

*FocalLoss* is used to calculate the loss value of a mask. Since the ratio between categories in a remote sensing dataset is very unbalanced, using cross entropy loss will cause the training process to be skewed towards the side with more categories. *FocalLoss* adds a modulating factor, $\gamma$, to overcome this drawback based on *CELoss*. The formula is as

follows, where $p$ is the predicted value, $y$ is the ground truth, and in this paper $\alpha = 0.25$, and $\gamma = 2$:

$$FocalLoss = \begin{cases} -\alpha(1-\alpha)^{\gamma}\log(p), & y = 1 \\ -(1-\alpha)p^{\gamma}\log(1-p), & y = 0 \end{cases} \tag{8}$$

*DiceLoss* [42] is derived from the dice coefficient, which is an ensemble similar to the measure function. *DiceLoss* is used as a measure function to evaluate the similarity between two samples and is designed to cope with a scenario of a strong imbalance between positive and negative samples in semantic segmentation. It is defined in the formula below, where $\varepsilon$ is used to prevent the extreme case where the denominator is 0. In this paper, $\varepsilon = 1$.

$$DiceLoss = 1 - \frac{2yp + \varepsilon}{y^2 + p^2 + \varepsilon} \tag{9}$$

## 4. Experiment

### 4.1. Dataset

In this article, we use the ISPRS Potsdam dataset to evaluate the performance of ACTNet. The ISPRS Potsdam dataset is extracted from the Potsdam region and contains 38 true radiographic images of $6000 \times 6000$ size. Each remote sensing image area covers the same size. Categories include Impervious surfaces, Buildings, Low vegetation, Tree, Car, and Clutter/background. The Clutter/background class includes bodies of water and other objects that look very different from everything else. Considering the size of the HRRS images and the limitations of GPU memory, we cut the images and corresponding labels into $600 \times 600$ pixel-sized images and then randomly divided them, with 80% going into a training set and 20% going into a validation set in disorder.

### 4.2. Evaluation Metrics

The semantic segmentation evaluation metrics used in this experiment contain two main categories. One is the metrics used to evaluate the accuracy of the model, including mean Intersection over Union ($mIoU$) and mean (class) accuracy ($mAcc$). The other category is a metric used to evaluate the inference speed and training speed of the module. Consider $mIoU$ as the primary metric, which calculates the intersection ratio of two sets and is widely used in semantic segmentation model evaluation. The formulas for the evaluation metrics are as follows:

$$mIoU = \frac{1}{k+1} \sum_{i=0}^{k} \frac{TP}{FN + FP + TP} \tag{10}$$

$$mAcc = \frac{1}{k+1} \sum_{i=0}^{k} \frac{TP}{FP + TP} \tag{11}$$

### 4.3. Implementation Details

We built our model using the MMSegmentation framework with Python 3.8. MMSegmentation is a deep-learning framework based on Pytorch, but is easier to scale and build complex networks with than the latter. To initialize our network parameters, the weights pre-trained by the BEiT [43] model on the ADE20K dataset were used. ResAttn and LFE modules use random initialization methods for the initial parameters, while the deformer block and Transformer decoder modules use the Kaiming initialization method [44] for the initial parameters.

For the hyperparameter setting we used a batch size of two and an initial learning rate of $1 \times 10^{-4}$. A warmup training strategy was used to avoid instability during training and to optimize the overall training effect. We used AdamW as the parameter update algorithm and Poly as the learning-rate adjustment strategy. All the experiments were trained in parallel on an NVIDIA Geforce RTX2080Ti with an 11-GB memory GPU and a maximum Epoch of 100. In addition, we used random crop, random flip, and other measures to enhance the training data.

*4.4. Comparative Experiments*

　　The ACTNet model was compared to other mainstream remote sensing semantic segmentation networks, namely the CNN-based FCN [21], U-Net [25], DeepLabV3+ [31], the Transformer-based Swin-ViT [19], ST-UNet [20], and SwinSUNet [33], respectively, on the Potsdam dataset using the same experimental settings, and the experimental results are shown in Table 2.

**Table 2.** Comparative experimental result on the Potsdam dataset.

| Method | Evaluation Metrics | | Inference Time (ms) | Training Time (min/epoch) |
|---|---|---|---|---|
| | *mIoU* (%) | *mAcc* (%) | | |
| FCN | 75.85 | 86.33 | 5.7 | 4.04 |
| U-Net | 77.23 | 87.45 | 8.5 | 7.31 |
| DeepLabV3+ | 78.47 | 88.12 | 13.83 | 9.87 |
| Swin-ViT | 79.63 | 87.73 | 27.93 | 15.60 |
| ST-UNet | 75.84 | 85.26 | 30.39 | 16.21 |
| SwinSUNet | 82.36 | 91.94 | 51.04 | 27.83 |
| **ACTNet** | **82.15** | **90.28** | **46.29** | **19.02** |

　　Our proposed ACTNet achieved an 82.15% in *mIoU* score and a 90.28% in *mAcc* score. The experiments showed that our model performed better than the CNN-based models or the Transformer-based models and required less training time and inference time compared to other CNN and Transformer-combined models

　　The visualization results from the comparison experiments are presented in Figure 7, where row (a) is the randomly selected image for the experiment and row (b) is the image corresponding to the ground truth value. From rows (c–e) of the figure, it can be seen that the traditional CNN-based model could not depict the specific outline of the object well due to too many details being lost during down-sampling, which resulted in less detailed results when performing multi-branching objectives. In column (1), the classifications of "Low Vegetation" and "Clutter/background" were incorrectly mixed due to the similarity of their colors. In column (2), the DeepLabV3+ model incorrectly split "Tree" into "Low Vegetation" and "Clutter/Background." The Swin-ViT model correctly classified these, but the area was incomplete. From the black box of columns (3) and (4), the ST-UNet and SwinSUNet were less effective in segmenting the "Clutter/Background" and "Building" objects due to foreground occlusion. The ACTNet achieved better results than the Transformer-based model due to the LFE module's ability to extract local features and its use of the mask-based segmentation method. ACTNet also outperformed DeepLabV3+, U-Net, and other CNN-based networks due to the global modeling capability of the attention mechanism. Furthermore, ACTNet also demonstrated better results on fragmented targets such as "Car" when compared to the CNN and Swin-ViT models.

　　Although the overall performance of ACTNet was superior to that of the other models, there is still potential for improvement regarding the segmentation effect. We analyzed the test results and visualized the confusion matrix, as shown in Figure 8. In the confusion matrix, we found that "Tree" was misclassified as "Building" or "Low Vegetation" in a large number of cases, which led to a decrease in the overall *mIoU* and *mAcc* values.

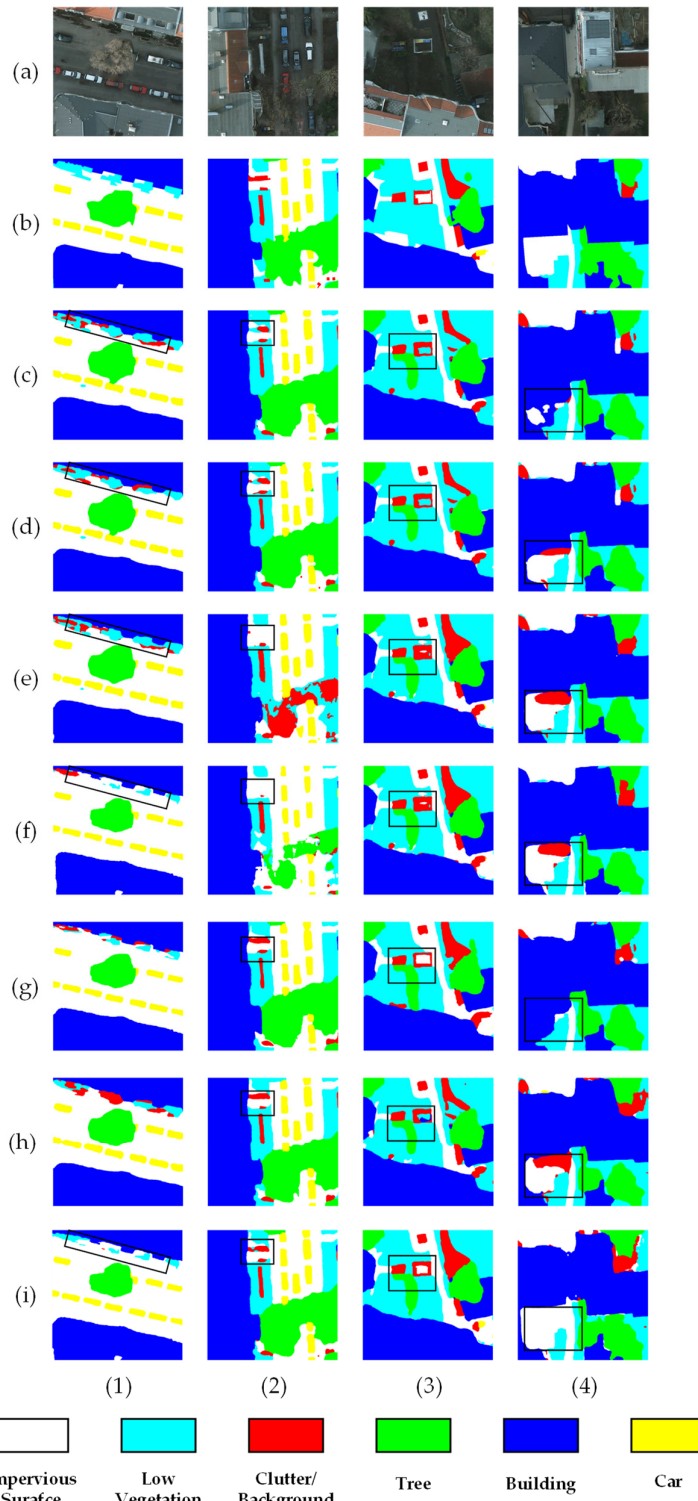

**Figure 7.** Comparative experimental results from the different models. The black boxes mark the areas with significant differences. Column (1–4) represents the segmentation results of four different test images. Row (**a**) represents the randomly selected image, row (**b**) represents the ground truth corresponding to the image, and rows (**c–i**) represent the experimental results from the FCN, U-Net, DeepLabV3+, Swin-ViT, ST-UNet, SwinSUNet, and ACTNet methods, respectively.

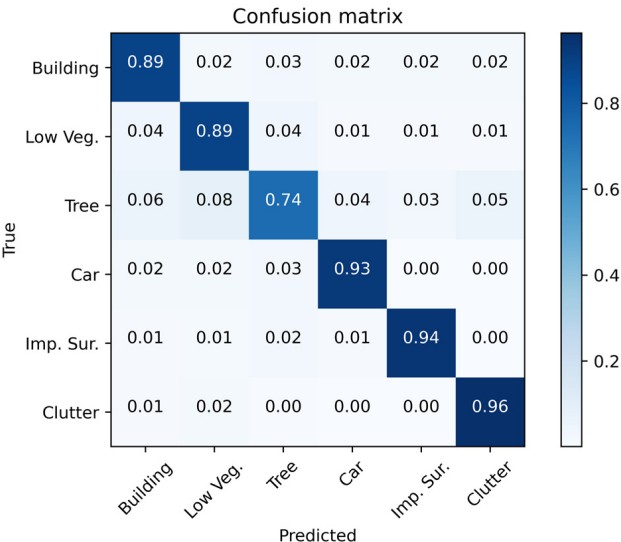

**Figure 8.** Confusion matrix for the ACTNet segmentation results.

### 4.5. Ablation Study

To further investigate the performance of the ResAttn, LFE module and the improved mask-based segmentation method in this paper, we conducted a series of ablation experiments on the Potsdam dataset. In these experiments, the baseline Swin Transformer model used a Swin-T configuration with layer numbers = {2, 2, 6, 2} and window size $M = 7$; all the models except ACTNet uniformly used Mask2Former as the segmentation decoder. Our image enhancement methods on these two datasets used random cropping and a 50% probability random flip, and the image resolution was uniformly scaled to $512 \times 512$ pixels. The overall experimental results are shown in Table 2, the specific values for each classification are shown in Table 3, and the visualization results of the ablation experiments are shown in Figure 9.

In Table 3, with the addition of the LFE and ResAttn modules leading to an increase in model's computations, the training time per epoch increased by 2.25 min and 3.13 min, respectively. In the experiments where both LFE and ResAttn were added, we used the pretrained parameters from the experiments with only the LFE module added. We froze the weights of the Swin Transformer backbone and LFE modules and updated the weights of the ResAttn module and decoder part. The experimental results showed only a small increase in the training time and an improvement in segmentation performance, which proves the effectiveness of an adapter in HRRS image segmentation. Meanwhile, the inference time increased by 18.36 ms compared to the baseline; however, this is acceptable in consideration of the improvement in classification accuracy.

**Table 3.** Overview of the results from the ablation experiments.

| Method | Evaluation Metrics | | Inference Time (ms) | Training Time (min/epoch) |
|---|---|---|---|---|
| | *mIoU* (%) | *mAcc* (%) | | |
| Swin-ViT | 79.63 | 87.73 | 27.93 | 15.60 |
| +LFE | 80.73 | 88.88 | 33.44 | 17.85 |
| +ResAttn | 80.38 | 88.32 | 36.05 | 18.73 |
| +LFE, ResAttn | 81.52 | 89.57 | 46.08 | 19.02 |
| **ACTNet** | **82.15** | **90.28** | **46.29** | **19.02** |

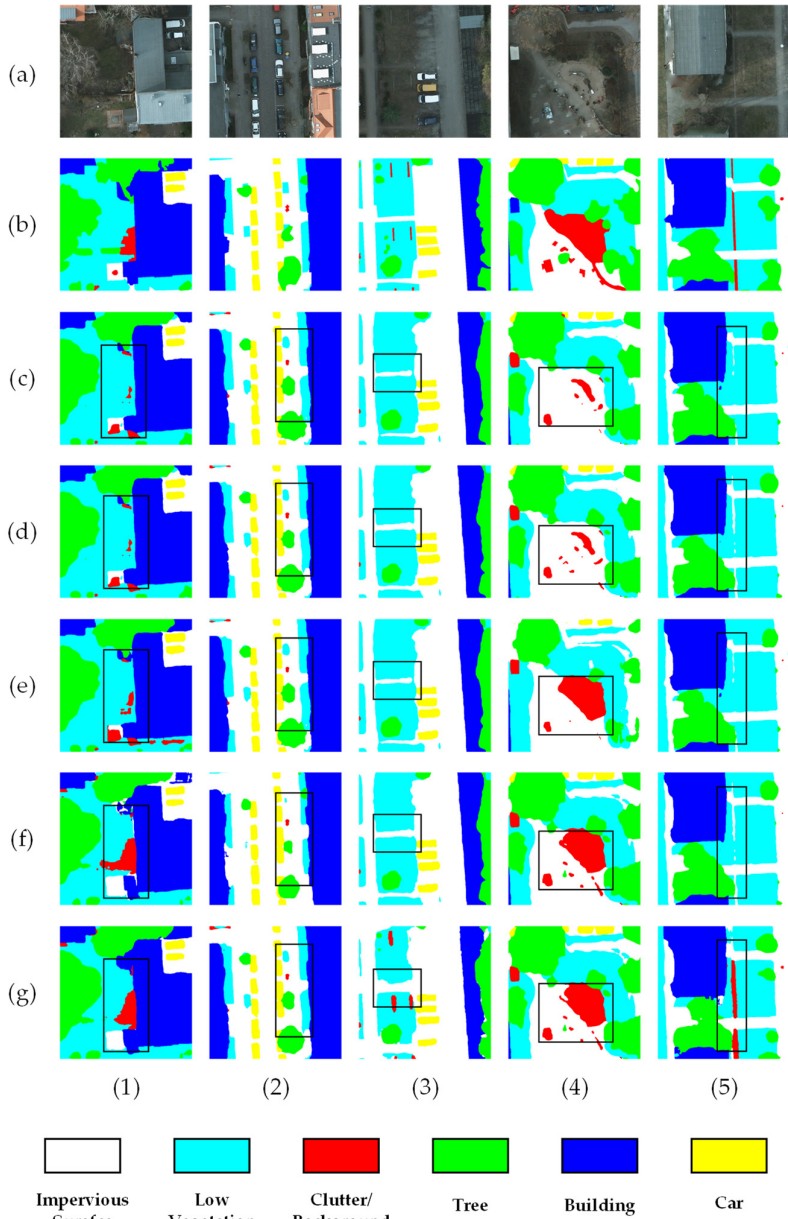

**Figure 9.** Visualization results from the ablation experiment. The black boxes mark the areas with significant differences. Columns (1–5) represent the segmentation results of five different test images. Row (**a**) represents the randomly selected image, row (**b**) represents the ground truth corresponding to the image, and rows (**c**–**g**) represent the experimental results from the Swin-ViT, Swin-ViT with LFE, Swin-ViT with ResAttn, Swin-ViT with LFE and ResAttn, and ACTNet models, respectively.

From Table 4, it can be seen that adding the LFE module to Swin-ViT increased the *mIou* value by 1.1% and the *mAcc* value by 1.15%. The IoU value for the classes "Low Vegetation", "Tree", "Car", and "Impervious Surface" were significantly improved. From Figure 9, row (d) shows a significant increase in IoU for the classes "Low Vegetation" and "Car". This demonstrates the effectiveness of CNN and a multi-scale structure in LFE modules for target edge analysis and small target segmentation. The relatively small increase in inference time relative to the improved segmentation effect is shown in Table 2, which proves the efficiency of the LFE module; these effectively compensate for the shortcomings of the Transformer model in this regard.

**Table 4.** Specific results from the ablation experiments.

| Method | IoU | | | | | | Evaluation Metrics | |
| --- | --- | --- | --- | --- | --- | --- | --- | --- |
| | Building | Low Vegetation | Tree | Car | Impervious Surface | Clutter/ Background | mIoU | mAcc |
| Swin-ViT(baseline) | 77.06 | 76.40 | 60.16 | 83.89 | 86.86 | 93.41 | 79.63 | 87.73 |
| +LFE | 77.74 | 77.77 | 61.20 | 85.00 | 87.78 | 94.89 | 80.73 | 88.88 |
| +ResAttn | 77.22 | 77.48 | 61.02 | 84.81 | 87.08 | 90.83 | 79.74 | 88.32 |
| +LFE, ResAttn | 78.13 | 78.33 | 63.79 | 85.20 | 88.26 | 95.41 | 81.52 | 89.57 |
| **ACTNet** | **78.19** | **78.54** | **65.38** | **86.09** | **88.89** | **95.81** | **82.15** | **90.28** |

Meanwhile, Table 4 shows that the *IoU* values of "Low Vegetation", "Tree", and "Car" obviously improved after adding the ResAttn module to Swin-ViT. This shows that the double attention mechanism in the ResAttn module and the fusion of token and query between the different features improved the segmentation effect for multiple identical targets in a certain region. However, the *IoU* value of "Background" decreased, and it can be seen from the black box of row (e) in Figure 9 that the "Impervious Surface" area had some incorrect segmentation as "Background". This indicates that the overuse of the attention mechanism caused the model to forcibly associate a target with other targets of different categories in a certain region, leading to segmentation errors. However, when the LFE module was used in combination with the ResAttn module, it could suppress some of the over-association effects of global modeling. As shown in Table 4, the use of both LFE and ResAttn improved *mIoU* by 1.89% and *mAcc* by 1.15%, with a significant improvements in all categories.

From the black box of row (f) in Figure 9, we can see that the misclassification of "Impervious Surface" and "Low Vegetation" was suppressed; the boundary between different targets was more clearly segmented. The segmentation results of the "Background" category were also closer to the ground truth. This is because after concatenating the output of LFE and ResAttn in the encoder part, the feature map set contained rich global modeling information and local feature information simultaneously, which further improved the model's ability to discriminate between object features. Finally, after the addition of our modified mask2former-based decoder under the above conditions, the model performance was further improved, which demonstrates the importance of fusing more high-level feature maps into the feature maps in multi-target segmentation.

## 5. Conclusions

In this paper, a high-performance HRRS image semantic segmentation method ACT-Net was proposed. To address the problem of the high computational complexity of the existing Transformer models for training downstream tasks and its dependence on a pre-training weight of large datasets, we proposed a Transformer-based adapter module for HRRS image semantic segmentation (ResAttn). This module uses a dual-attention mechanism to ensure the acquisition of global information while the structure of Swin-ViT remains unchanged. To enhance the extraction of edge and texture features, we designed a CNN-based LFE module and used a pyramid-like structure to fit multi-scale objects. Moreover, we used a mask-based segmentation method with a residual-enhanced deformable attention block to further improve the extraction of small objects. Our series of experiments on the Potsdam dataset demonstrated the excellent performance of ACTNet. In the future, we hope to further reduce the overall training parameters and computational resources used by ACTNet. We will try to find a unified semantic segmentation network based on the structure of ACTNet to support more HRRS image datasets. Furthermore, we will explore its role in urban rail transportation planning, and to demonstrate the generality of the ACTNet structure.

**Author Contributions:** Conceptualization, Z.Z. and F.L.; methodology, Z.Z. and F.L.; software, F.L.; validation, Z.Z.; formal analysis, C.L.; investigation, Q.T. and H.Q.; resources, Q.T.; data curation, F.L.; writing, Z.Z. and F.L.; original draft preparation, F.L.; visualization, Z.Z.; supervision, F.L. and Q.T.; project administration, C.L. and H.Q. All authors have read and agreed to the published version of the manuscript.

**Funding:** This research was funded by the National Key Research and Development Program under Ministry of Science and Technology of the People's Republic of China (2020YFB1600702).

**Data Availability Statement:** Not applicable.

**Conflicts of Interest:** The authors declare there are no conflict of interest.

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
