# Peer review of "ACTNet: A Dual-Attention Adapter with a CNN-Transformer Network for the Semantic Segmentation of Remote Sensing Imagery"

_remotesensing, doi:10.3390/rs15092363_

Round 1

Reviewer 1 Report

In this paper, the authors presented a unique model i.e, ACTNet:A Dual-Attention Adapter with CNN-Transformer Network for Semantic Segmentation on Remote Sensing Imagery

Therefore, it is interesting and attractive. However, it should be major revised to enhance the quality, as follows:

1) In Section 1, authors should make three sub sections, motivation, contributions and organization of the paper

2) Literature review is not up to the mark, Pl add in section 2

3) At the final end of section 1, the authors should indicate the rest of this paper is organized how.

4) Contributions of the research paper is missing or its not clearly mentioned

5) A summary table of should be provided for convenience for the readers in literature review section.

6) Eq.1 to 4 need to represented well

7) Figure 5 should be re-presented. Moreover, all the parameters should be explain clearly .

8) All Figures should be enhanced at the high resolution

9) A comparison table must be there with other existing model

10) Finally, the authors should double-check all formation, typos, and writing throughout the paper.

Reviewer 2 Report

In recent years, the Transformer model achieve the state of art performance in remote sensing image semantic segmentation. Nevertheless, the Transformer model's high computational complexity and dependence of pretraining weight on large datasets lead to slow convergence during the training. This paper proposed a novel adapter module (ResAttn) and a CNN-based LFE module to improve the training speed and feature extraction ability.

To enhance this work, I have the following questions and comments:

1. The main contributions of this paper should be summarized at the end of the section “Introduction” for better legibility.

2. You mentioned that the adapter module (ResAttn) is able to improve the model training speed, so I believe that you should compare the training time of different models to verify it. The effect on the training speed of your LFE module and Deformer Block should also be discussed in the paper.

3. In the section “Experiment”, you compared the ACTNet with the conventional semantic segmentation method, FCN, U-Net, DeepLabV3+, and Swin-VIT. However, many variants of these methods have been developed in recent years. so I believe you should compare ACTNet with more state-of-the-art approaches, especially the variants of Swin-VIT.

Reviewer 3 Report

The authors present a fascinating paper. Despite this, the paper needs improvement. I added a few suggestions.

Sections 2.1, 2.2, and 2.3 are like subsections of a literature review section or part of the introduction. Check section 2.

Methodology section. In the introduction sentence, add more content; for example, mention the objective of the new method developed.

In Figure 2, I suggest marking the encoder and decoder parts to help the reader understand the method.

Overview section. Figure 2 shows the method takes an image as input; then, a stem block process the image. But the overview section explains the LFE module, which goes after the stem block. Check the overview section to follow the Figure 2 logic; I suggest changing Figure 2 or adding a new figure that illustrates the explanation of the 3.1 section.  

Overview section. Which is the size of the encoder output? What is the size of the convolution layers? What is the size of the Deformed block layers?

3.2 ResAttn. As shown in Figure 2, the first step is the stem block. I suggest starting by explaining the stem block in detail, then explaining the ResAttn.

The first time the LFE is mentioned needs to be defined (Local Feature Extractor) no later in the paper.

Where does the Local Feature Extractor occur in Figure 2?

I suggest re-writing the Methodology section. The section is not easy to follow since it needs coherence. Also, the subsection topics are not in the same order as the main method's block illustrates in Figure 2. Likewise, a few subsections' topics need to be illustrated in Figure 2. Finally, the methodology section needs to be rewritten so the readers can follow if they want to replicate the methods explained in the paper. 

I suggest adding a confusion matrix to help illustrate the results obtained.

Figure 6. I suggest adding a legend with the categories and their colors.

Section 4.4 Comparative Experiments. I suggest reviewing the section. 

A more detailed comparison would help better illustrate the performance of the proposed method. Also, add a comparison between the result of each category (the comparison of the average result is not enough).   

Figure 7. It is missing a detailed explanation of the area in the black box.

Introduction and Conclusion. I suggest adding the category that the method identifies.

Round 2

Reviewer 1 Report

Authors are addressed all the queries.

Reviewer 3 Report

The authors addressed my comments.